# Biomass Estimation of Urban Forests Using LiDAR and High-Resolution Aerial Imagery in Athens–Clarke County, GA

**Katrina Ariel Henn** * and **Alicia Peduzzi** *

Warnell School of Forestry and Natural Resources, University of Georgia, Athens, GA 30605, USA
* Correspondence: katrina.henn@uga.edu (K.A.H.); apeduzzi@uga.edu (A.P.)

**Abstract:** The benefits and services of urban forests are becoming increasingly well documented, with carbon storage being the main focus of attention. Recent efforts in urban remote sensing have incorporated additional data such as LiDAR data but have been limited to sections of an urban area or only certain species. Existing models are not generalizable to remaining unmeasured urban trees. To make a generalizable individual urban tree model, we used metrics from NAIP aerial imagery and NOAA and USGS LiDAR data for 2013 and 2019, and two crown-level urban tree biomass models were developed. We ran a LASSO regression, which selected the best variables for the biomass model, followed by a 10-fold cross-validation. The 2013 model had an adjusted $R^2$ value of 0.85 and an RMSE of 1797 kg, whereas the 2019 model had an adjusted $R^2$ value of 0.87 and an RMSE of 1444 kg. The 2019 model was then applied to the rest of the unsampled trees to estimate the total biomass and total carbon stored for all the trees in the county. Recommendations include changes to ground inventory techniques to adapt to the current methods and limitations of remote sensing biomass estimation.

**Keywords:** urban tree biomass; carbon storage; LiDAR; urban remote sensing; individual crown segmentation; LASSO

## 1. Introduction

The literature has increasingly documented the numerous ecosystem services and benefits of urban forests. These range from ecological to social services. On the social front, urban trees have been found to increase mental health, decrease hospitalizations, and decrease the rate of sickness. Ecologically, urban trees provide wildlife habitat, shade, water mitigation, and decreased soil erosion [1]. Increasing the number of these benefits is quickly becoming a central focus, as more people migrate to expanding cities and rural woodlands are cut down.

Perhaps one of the most significant services that is currently receiving great attention is the urban forest's capacity to store carbon. A recent study by the US Department of Agriculture (USDA) and the US Forest Service (USFS) reported that, in terms of terrestrial carbon storage, all trees, including urban trees, were collectively the largest carbon sink in 2019 [2]. They offset more than 11% of total greenhouse gas emissions in 2019. The topic of carbon storage has become even more urgent with the United Nations Intergovernmental Panel on Climate Change 2022 report, which issued a dire need to cut greenhouse gas emissions [3]. Therefore, it will be important for cities to know the contributions of their urban forests toward carbon storage, and how this might change in the future based on management decisions.

Several methods of estimating tree biomass, carbon storage, and sequestration have been developed to address this need. Common methods include relating plot-level inventory data to satellite or aerial imagery and extrapolating to unsampled canopy area [4,5]. The variables related to the inventory data may be physical (if a digital elevation model is available), spectral, or image texture or shape. The canopy area itself may be determined

using vegetation indices, as well as pixel-based and object-oriented classification [6]. Traditional forest plot inventory facilitates the categorization of biomass in this way, and the increased frequency of forest inventory such as Forest Inventory and Analysis (FIA) plots allows for increased monitoring over time [7,8].

Perhaps the most popular software among urban foresters that aims to fill in the gap of relating urban plot data to remote sensing data—currently aerial imagery—is the iTree software suite, specifically the Urban Forest Effects (UFORE) Model in the iTree Eco software, developed by the USFS [9]. The software uses ground inventory data and remotely sensed imagery to estimate the total benefits across the region of interest. It also represents the benefits by translating them into financial terms. Requiring only plot-level data as input, the iTree software has been a useful tool for urban foresters to promote forest benefits and its condition to the municipality and public. Its calculation method involves multiplying plot-level data across all the area in the image that is classified as forest. This basic method generalizes across the rest of the unmeasured forestry area using only the plot data given. This results in lost details and uncaptured spatial heterogeneity across the urban landscape.

Limitations from image-only methods such as those used by iTree, as mentioned above, can be overcome with the integration of additional three-dimensional data. For example, researchers have explored the relationship between biomass and various vegetation indices, especially the normalized difference vegetation index (NDVI), which is a common indicator of vegetation health and therefore to an extent growth, from aerial or satellite imagery. While there is a modest correlation, it is vastly improved by adding light detection and ranging (LiDAR) metrics [10–14]. For example, one study found that the combination of LiDAR and multispectral imagery metrics produced better results for biomass, basal area, and volume estimates in mixed-species forests [15]. Another showed improved species identification by using LiDAR and multispectral imagery [16]. Therefore, such methods that rely only on imagery and its two-dimensional derivations are constrained by the lack of three-dimensional data, which would enable more accurate identification of trees, their crowns, and various attributes such as height and crown density.

The increasing presence of LiDAR data provides this three-dimensional aspect and has major potential for urban forestry applications. Used together with satellite or aerial imagery in a process called data fusion, LiDAR has been implemented in research to identify urban tree species, calculate leaf area, track shifting understory plant species, identify and delineate individual tree tops and crowns, and estimate biomass [13,17–20]. Such applications are promising not only because of the additional detail added to urban tree inventories but because of the time and labor saved that would otherwise be required for an equivalent ground inventory. They also apply to areas that urban foresters might find more difficult to reach, such as private property. Finally, the LiDAR data will have other urban planning and development uses outside of urban forestry. The investment into one LiDAR collection flight would apply to many uses, such as urban land cover classification, urban feature extraction, building object detection, and automated urban driving [21–23]. These benefits are especially appealing during tight municipal budgets and understaffed departments.

The products (canopy cover, species, biomass, etc.) from these applications aid urban forest management decision making to reach or maintain multiple goals, and mapping their spatial distribution enables an easier assessment of unevenness. This is important for environmental equity, a topic that has received considerable attention because the unevenness of tree distribution can imply unevenness in the services and benefits associated with those tree traits. Initial urban mapping attempts showed the distribution of canopy cover across different census block groups or tracts based on race or income [24–26]. More recently, attention has turned toward mapping biomass values across neighborhoods and using land green space value by incorporating multiple values of vegetation. In addition, biomass is associated with the leaf area index (LAI), which is the value of the leaf area of a tree over a ground surface area [27]. A higher LAI indicates greater leaf area, which generally

means higher photosynthetic rates and other ecophysiological processes. Thus, the LAI has been linked to increased benefits and services but is challenging to estimate using remote sensing data [28,29]. Integrating biomass as an alternative or additional variable into urban green indices would provide a more complete picture for urban decision makers. Understanding biomass distribution in addition to overall canopy cover would enhance inventory knowledge and distribution of benefits and services across the urban environment.

Therefore, this study aims to (1) generate a model using LiDAR data and aerial imagery to calculate biomass across all trees in a county by using regression with LiDAR metrics and (2) assess and visualize the urban forest structure and distribution of the biomass, carbon, and other urban tree attributes. This research further aims to serve as a guide to future larger-scale urban tree biomass estimation projects.

## 2. Materials and Methods

### 2.1. Location

Athens–Clarke County (ACC) is a consolidated city–county government, located about 110 km (70 miles) northeast of downtown Atlanta, Georgia, in the southeastern United States (Figure 1).

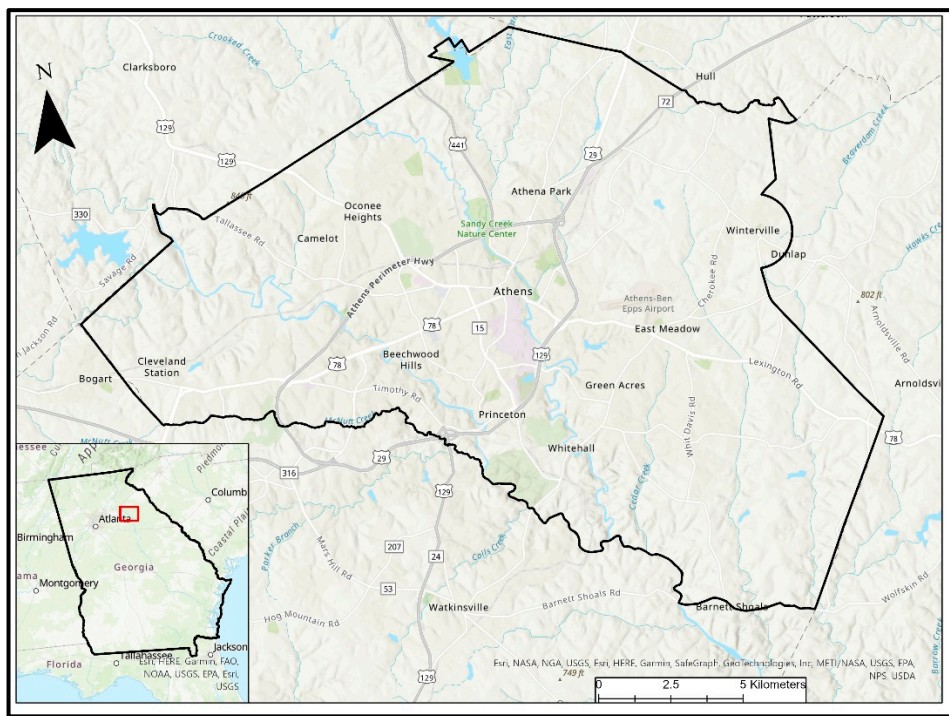

**Figure 1.** Athens–Clarke County (red box) in the state of Georgia, United States.

Athens is approximately 122 square miles and has the smallest land area of Georgia's 159 counties. One of the highest points in Athens is City Hall, located at 761 feet above sea level [30]. Athens has a humid subtropical climate. Its climate is known for hot summers, with precipitation being consistently high throughout the year. The average high in the summer reaches over 90 degrees Fahrenheit.

### 2.2. Field Data Collection

To randomize the right-of-way (ROW) and public tree points, a county street shapefile from Athens–Clarke County was used and buffered to about 9 m (30 ft), as based on the GIS inspection of the average neighborhood street and ROW area. Park spaces were added to this as well. Tree points were spawned inside the buffer and park boundaries and then clipped to the outside of the parcel data polygons to ensure the points were all outside of



the private property. A series of randomly selected ROW trees from the Athens–Clarke County inventory were also added. Finally, the sample points that were located in places with no trees were eliminated. The points were then downloaded as a kmz file to use for navigation.

Field data collection was carried out in the summer of 2021, during which the points were visited, and tree data were collected. The nearest obtainable ROW tree to the point was selected. In total, 560 ROW trees across the county were measured for the diameter at breast height (dbh) using a diameter tape and height using a Nikon Forestry Pro hypsometer. The diameter at breast height was measured upslope at 1.37 m (4.5 ft) above the ground. The instruments were used by the same person during the whole process to avoid mixing user bias. In arriving at each point, the nearest ROW tree was chosen. While determining ROW or public trees is seldom a clear-cut process, it was made easier through the previous process of clipping to outside the parcel data, as well as using utility markers on-site as guides. Following iTree guidelines, each stem of a multistemmed tree was measured. A Trimble Juno T41 GPS (2–4 m accuracy) was placed at the base of each tree, which allowed us to average the location data for the entirety of the data collection process, ensuring more accurate tree points. If possible, tree species, or genera if not species, were recorded. Tree data were collected from a variety of conditions, namely in more forested conditions around other trees or as a more isolated tree (Figure 2). Any tree with no overlapping of tree crowns was classified as isolated. Trees overlapping with each other were classified as growing in woody conditions.

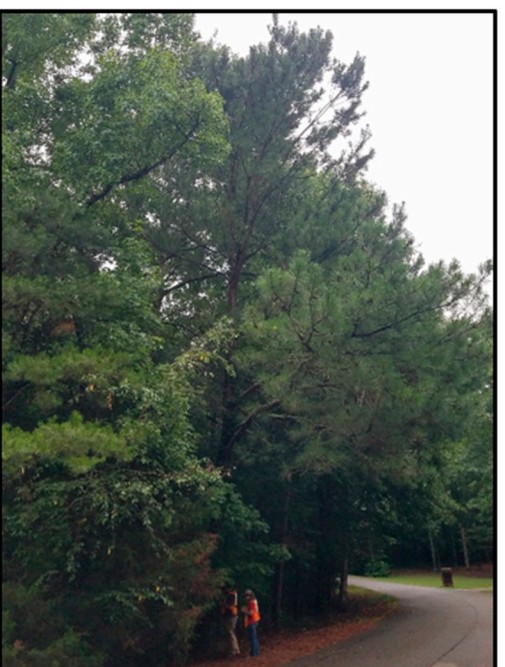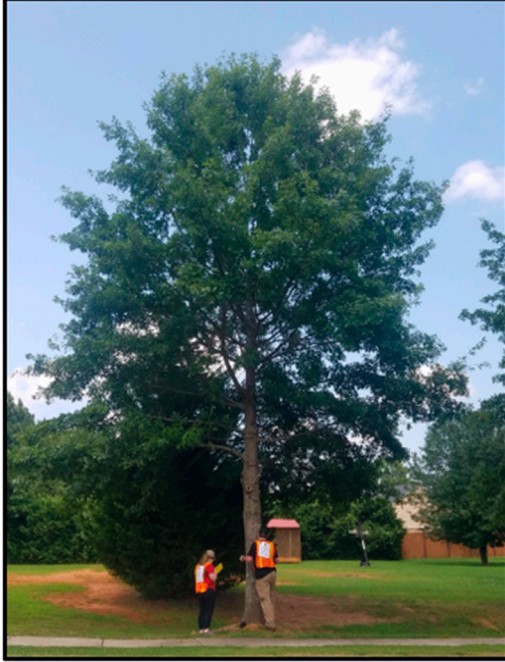

**Figure 2.** Woody conditions (**left**) and isolated conditions (**right**) for tree data collected in Athens–Clarke County, Georgia, US.

### 2.3. LiDAR and Image Processing

Four-band color near-infrared (CNIR) 1 and 0.6 m aerial imagery data from 2013 and 2019 were acquired from USGS Earth Explorer. The aerial imagery was acquired from the USDA. The CNIR characteristics allowed for four bands of data: red, green, blue, and near-infrared. Because NDVI is a well-known indicator of vegetation health and uses only red and near-infrared wavelengths. NDVI was able to be calculated for all urban trees. NDVI, first proposed by Kriegler [31], was calculated using the following equation:

$$\frac{NIR - red}{NIR + red} \tag{1}$$

Additional ratios and formulas using the CNIR bands were calculated, following Machala and Zejdová [32] (Table 1).

**Table 1.** All variables calculated per tree crown from four-band aerial imagery collected in Athens–Clarke County, Georgia, US. The * indicates multiplication.

| Name | Formula |
| --- | --- |
| OSAVI | $\frac{NIR-red}{NIR+red+0.16}$ |
| Four bands ratio | $\frac{NIR_{mean}+red_{mean}}{blue_{mean}+green_{mean}}$ |
| Ratio 1 | $\frac{NIR_{mean}}{red_{mean}} + \frac{green_{mean}}{blue_{mean}} * \frac{CHM_{mean}}{DSM_{sd}}$ |
| Ratio 2 | (red_sd * NIR_mean)/(blue_sd + green_sd) |
| NIR to blue ratio | NIR_mean/blue_mean |
| Three-band ratio | NIR_mean/red_mean/green_mean |
| Ratio 3 | (NIR_sd * NIR_mean)/(blue_sd * blue_mean)—(green_sd * green_mean)/(red_sd * red_mean) |
| CHM * DSM st dev. | CHM_mean * DSM_sd |
| NDVI * NIR sd | NDVI_mean * NIR_sd |
| NDVI * DSM sd | NDVI_mean * DSM_sd |
| NIR to red sd | NIR_sd/red_sd |
| NIR * DSM sd | NIR_sd * DSM_sd |
| Three sd ratio | (NIR_sd * DSM_sd)/red_sd |
| Ratio 4 | (NIR_sd * DSM_sd)/red_sd * NDVI_mean * CHM_mean |
| Blue ratio | Blue_mean/(red_mean + green_mean + blue_mean) |
| NIR-to-red ratio | (NIR_mean * NIR_sd)/(red_mean * red_sd) |
| Green ratio | Green_mean/(red_mean + green_mean + blue_mean) |
| Ratio 5 | NIR_mean/NIR_sd * blue_ratio/red_mean |
| Ratio 6 | (NIR_mean—red_mean + green_mean + blue_mean)/(NIR_mean + red_mean + green_mean + blue_mean) * CHM_mean |
| Green-to-blue Ratio | Green_mean/blue_mean |

We used LiDAR data from 2013 originally collected by the NOAA (available at the Athens–Clarke County GIS Open Data website) [33] in 2.59 square-kilometer (1 square-mile) tiles. The 2019 LiDAR data were acquired from the USGS (available at LiDAR Explorer) in 1 square-kilometer (1,000,000 square-mile) tiles. In total, 364 tiles covering the county were used. The 2013 and 2019 data had a point density of about 1 point/m$^2$ and 4.7 points/m$^2$, respectively. There were up to four returns per pulse for the 2013 data and five for the 2019 data. Points were already classified as ground, overlap, and water for both.

The following preprocessing was used for both 2013 and 2019 LiDAR data. The LiDAR points were filtered to exclude noise. From the filtered LiDAR dataset, a digital elevation model (DEM), a digital surface model (DSM), and a canopy height model (CHM) were created [34] (Figure 3). The DEM was produced by first filtering for classification equal to 2 (ground) and then using Kriging with a 1 m resolution. The DSM was produced by subtracting the DEM from the cleaned LiDAR dataset. This process normalized the data. The DSM contained trees, power lines, towers, and buildings. To create the CHM, the OSAVI and NDVI values were extracted from each LiDAR point, and points with values under −0.15 and −0.30, respectively, were eliminated in order to create the true urban canopy height model. In addition, the points were classified as linear (power lines) or planar (roofs) in nature using the shape detection algorithms available in the lidR package

version 3.1 [35,36]. These points were removed. Finally, all points under about 2 m (6 ft) were removed in accordance with the estimated height of small structures and most cars.

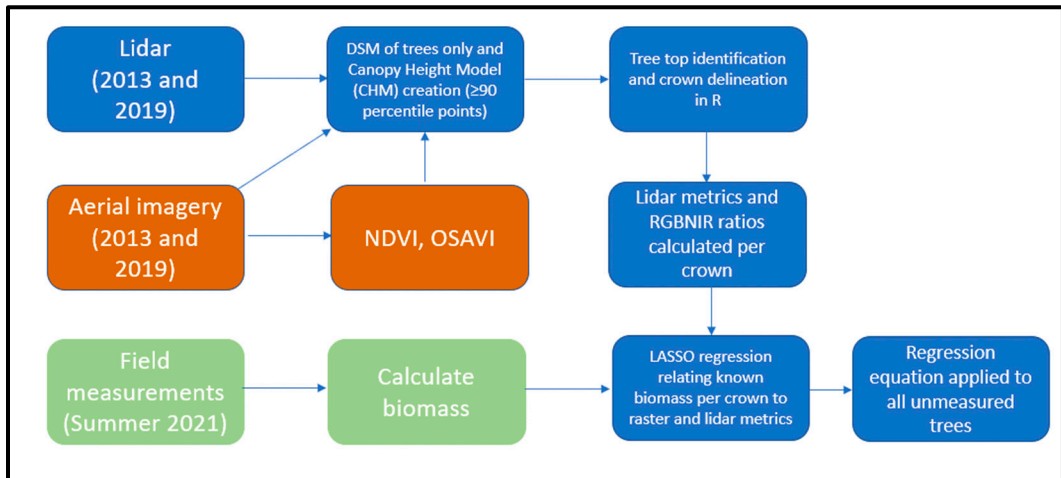

**Figure 3.** Workflow of data processing and biomass modeling.

Traditional LiDAR metrics were calculated for the field trees, using the *LASmetrics* function in the ForestTools library as a guide [37]. This calculated the standard LiDAR metrics, including the 1st percentile to the 99th percentile of height, median height, and the max and minimum heights, as well as the number of returns above the mean, mode, and other thresholds. The laser penetration index (LPI) and the laser interception index (LII) were also calculated [38,39]. The same series of LiDAR metrics were computed for all points above 1 m, followed by all points above 1.83 m (6 ft), and then for all points above 14.63 m (48 ft). Additionally, 1 m was chosen to accommodate low-lying crowns; 1.83 m was considered to eliminate taller obstacles, including parked cars under trees; and 14.63 m was the median height of all field trees. This proliferated as many potential variables as possible.

### 2.4. Treetop Detection and Crown Delineation

Using the filtered and normalized LiDAR data, a local maximum filter from the *locate_trees* function in the lidR library was used. The point feature output was input into the *silva2016* function in the lidR package to delineate crowns. The *silva2016* function is based on a seed and Voronoi tessellation method [40]. Multiple other treetop and crown delineation techniques were attempted in both R and Ecognition. Specifically, treetop seeds were either from the local maximum filter from *locate_trees* or the original GPS points. Other crown methods included watershed segmentation and *dalponte2016*, which is a seed and region-growing algorithm [41]. The results were visually compared over a variety of situations, including natural forested conditions, isolated street trees, and trees placed near buildings.

### 2.5. Biomass Calculations and Modeling for 2013 and 2019

In order to calculate biomass for all unmeasured trees, biomass was first calculated for all the trees measured in the field. To compute biomass for all trees except pine, allometric equations from the USDA Urban Tree Database and Allometric Equations [42] were used, along with the ROW tree dbh and height data collected from the field. The allometric equation for pine trees was used following Cieszewski et al. [43], who adapted it from Taras and Phillips [44]. This pine allometric equation replaced the one from the Urban Tree Database because (1) pines represented a large portion of the sample; (2) the equation was pine-specific instead of general conifer; and (3) the equation was localized to the southeastern US. When possible, species-specific volume or biomass equations were used

or, if available from the Urban Tree Database, were matched up with a similar species and its volume or allometric equation. If neither of these were available, the urban general broadleaf or conifer volume equations were used. If the equation's result was volume, it was multiplied by each species' dry-weight factor to obtain biomass [42,45,46]. Dry-weight factors were obtained from a variety of sources, and the Urban Tree Database, the Global Wood Density Database, the USFS Specific Gravity and Other Properties of Wood and Bark for 156 Tree Species Found in North America, the USFS Wood Handbook for Engineering, and the Wood Handbook were given priority, in that order (Table 2) [42,47–50]. The Urban Tree Database and its recommendations were given top priority since applying non-urban allometric equations has previously resulted in noticeable over- or underestimation and variability as high as 300% for individual trees [51]. Some factors were used at the genus level, either because that is what was available or because that was the finest level recorded in the field. For example, the general factor values for the white oak and red oak groups were used when the tree was only listed to the genus and group level. Biomass was calculated per stem, and the total biomass for a tree with more than one stem was the sum of each stem. Biomass was multiplied by 1.28 to incorporate belowground biomass [42]. Finally, because carbon is known to be roughly half of a tree's dry-weight biomass, each tree's biomass was multiplied by 0.5 to calculate carbon storage. The result was also multiplied by 3.67 to compute stored carbon dioxide ($CO_2$).

**Table 2.** References used for species' dry-weight factors for calculating biomass from field data obtained in Athens–Clarke County, Georgia, US.

| Reference (in Descending Order of Priority) | Species |
| --- | --- |
| Urban Tree Database and Allometric Equations | *Acer negundo, Acer saccharinum, Betula* spp., *Carya illinoinensis, Cercis canadensis, Cornus florida, Fagus grandifolia, Ginkgo biloba, Ilex* spp., *Juniperus virginiana, Lagerstroemia indica, Liquidambar styraciflua, Liriodendron tulipifera, Magnolia grandiflora, Morus rubra, other, Platanus occidentalis, Prunus caroliniana, Prunus serotine, Prunus serrulate, Quercus macrocarpa, Quercus nigra, Quercus phellos, Ulmus alata, Ulmus Americana, Ulmus parvifolia* |
| Global Wood Density Database | *Acer rubrum, Oxydendrum arboretum, Quercus falcata, Quercus michauxii, Quercus stellata* |
| Specific Gravity and Other Properties of Wood and Bark for 156 Tree Species Found in North America | *Albizia julibrissin, Catalpa bignonioides, Diospyros virginiana, Halesia diptera, Magnolia lilliflora, Magnolia virginiana, Quercus texana* |
| Wood Handbook: Wood as an Engineering Material | *Metasequoia glyptostroboides, Quercus acutissima, Quercus alba, Quercus rubra* |
| Wood! Identifying and Using Hundreds of Woods Worldwide | *Cryptomeria japonica, Cupressus × leylandii* |

Because the urban general broadleaf equation is known to overestimate biomass by up to 50% [42], urban species-specific biomass results were compared against the general broadleaf biomass results from the field-measured trees to determine if a factor should be used.

In order to generate the biomass models for 2013 and 2019, the relationship between tree biomass, LiDAR metrics, and spectral variables needed to be determined using the field tree data. First, for both years, the metrics needed to be extracted from each individual tree, which was accomplished using the crowns produced from the *silva2016* algorithm.

Next, we extracted the metrics for each crown polygon, as detailed in Section 2.3, resulting in over 300 variables. As the number of variables was greater than the final sample sizes, we needed to reduce the number of variables inputted into the model. We left as many variables as possible in order to maintain a square singular matrix for the regression

solution. That meant we selected the top 89 variables based on the correlation values sorted from highest to lowest, and for 2019, we selected 100 variables. Following variable selection to be inputted into the model, the Box–Cox method was used to investigate the data for normal distribution. Following the Box–Cox results, the data were transformed using the natural logarithm. Then, applying a least absolute shrinkage and selection operator (LASSO) regression as the most efficient method to model LiDAR data [52], the crown variables, multispectral variables, and LiDAR metrics most highly correlated with biomass were selected based on the best model, with the minimum lambda value. This was conducted on the overall field tree dataset and then on two subgroups: (1) one group containing trees that were mostly isolated from other trees and (2) another group containing trees that were in more forested or woody conditions. Crowns in each dataset were eventually filtered to keep only those whose LiDAR heights closely matched the measured field height. After this initial filtering, we used Google Street View to examine the remaining trees whose recorded heights were still largely different from the LiDAR heights. This was especially important for the 2013 dataset due to the larger span in time from the 2021 field data collection. Any tree that appeared to have noticeably grown in height since 2013 was removed from the sample. Then, potential outliers and influential trees were examined by filtering for any observations with standardized residuals greater than or equal to the absolute value of 2, and by filtering any observation having three times greater value than the mean using Cook's distance. We furthermore again visually inspected their accompanying field photo and/or Google Street View image, if available. The single best-performing final model based on adjusted $R^2$, RSME, and mean error (bias) was applied to predict the biomass for the remaining unmeasured tree crowns across the county. Then, 10-fold cross-validation or out-of-sample testing was used to evaluate the accuracy of the predictive model. This method randomly divided the dataset into training and testing sets and applied the model in ten iterations.

### 2.6. Mapping 2019 Biomass across Athens–Clarke County

Using the *silva2016* function paired with seeds from *locate_trees*, crowns were generated for every tree on a per-LiDAR tile basis. LiDAR metrics, crown variables, and spectral variables were generated for every crown. These variables were then used in the final 2019 model to predict biomass. The overall tile results were also summarized, such as total values and quantiles for the NDVI, height, crown area, biomass, carbon, and carbon dioxide. Carbon and carbon dioxide storage estimates were calculated following the procedure mentioned in Section 2.5. This process was run on the University of Georgia's Georgia Advanced Computing Resource Center's (GACRC) Sapelo2. Sapelo2 is a Linux (64-bit Centos 7) high-performance computing cluster. The script was run on an Intel Xeon processor and used 10 and then 30 cores simultaneously across 70 GB RAM via the *mc\** functions from the *parallel* R package [53].

The resulting biomass estimates were examined for distribution patterns based on both land ownership and land cover type per census block group. When determining land ownership, any tree points outside of the QPublic parcel data were considered ROW. Using a zoning shapefile layer from the ACC Open GIS Data website, the remaining trees were classified as private, park/recreational, or educational/government. For determining land cover data, the 2019 data from the National Land Cover database were used. Thus, biomass distribution totals and percentages were calculated per block group based on land ownership and land cover type. This allowed for an examination of each neighborhood's dependency on the most important urban tree canopy land managers for the area.

## 3. Results

### 3.1. Inventory

Five hundred and sixty ROW and public trees were measured across Athens–Clarke County (Figure 4). Of these, 161 were marked as isolated. Figure 5 shows the species collected, as well as the species volume or biomass equation with which it was matched.

Table 3 lists the descriptive statistics for the tree diameter and heights measured in the field, as well as the biomass and carbon calculated for each.

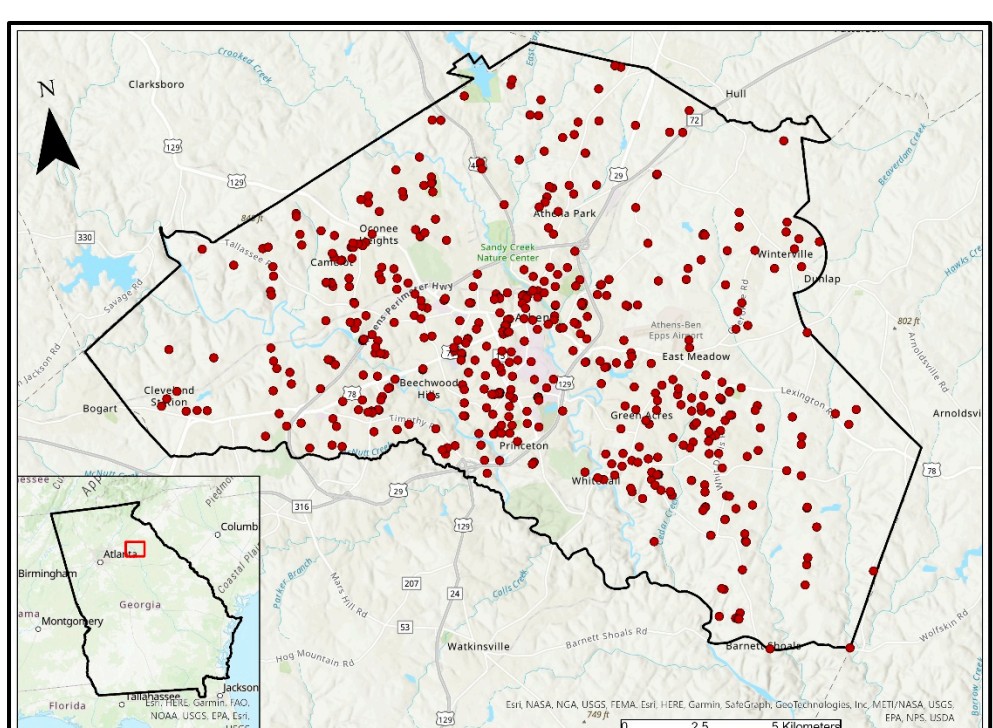

**Figure 4.** Location of 560 field tree GPS points in Athens–Clarke County (red box), Georgia, US.

**Table 3.** Descriptive statistics for diameters and heights measured in all field trees, as well as the biomass and carbon calculated for each (*n* = 560). Data were collected in Athens–Clarke County, Georgia, US.

|  | Min, Max | Average | Standard Deviation |
|---|---|---|---|
| **Tree diameter (cm)** | 3.8, 133.6 | 35.4 | 23.2 |
| **Tree height (m)** | 2.7, 42.7 | 15.1 | 7.8 |
| **Tree biomass (kg)** | 1.5, 21,678.4 | 1352.9 | 2484.5 |
| **Tree carbon (kg)** | 0.8, 10,839.2 | 676.4 | 1242.3 |

*3.2. Crown Segmentation*

The *silva2016* algorithm was chosen for both its performance and availability. Table 4 shows the crown sizes for the final 2013 and 2019 datasets using *silva2016*. When this method was paired with treetop seeds from the *locate_trees* algorithm, it showed better results than *dalponte2016* and than when using the GPS field tree points as the only tree-tops/seeds for either algorithm. In the latter method, the algorithms typically resulted in crown polygons of similar size across trees of different tree crown sizes. In addition, using the *dalponte2016* algorithm paired with the GPS points as seeds resulted in the fewest number of crowns generated, thus reducing sample size. With other treetops as seeds from the *locate_trees* function, the *silva2016* created crowns of various sizes that better matched their real counterparts. However, there were still issues of crown overflow in which a single crown covered multiple trees, especially in woody situations. Finally, no matter the method used, treetops and crowns sometimes covered building edges, despite the point filtering steps previously conducted. This created false crowns. Figure 6 shows an example of crown overflow over buildings and the false detection of a few crowns beside a roof. The crown boundaries appear to start properly on the side of the crown away from

the building but expand into the building area on the other side. This was likely due to difficulties with filtering LiDAR points to only vegetation. More detailed filtering, treetop detection, building delineation, and crown delineation algorithms are needed to address this challenge posed by the urban environment.

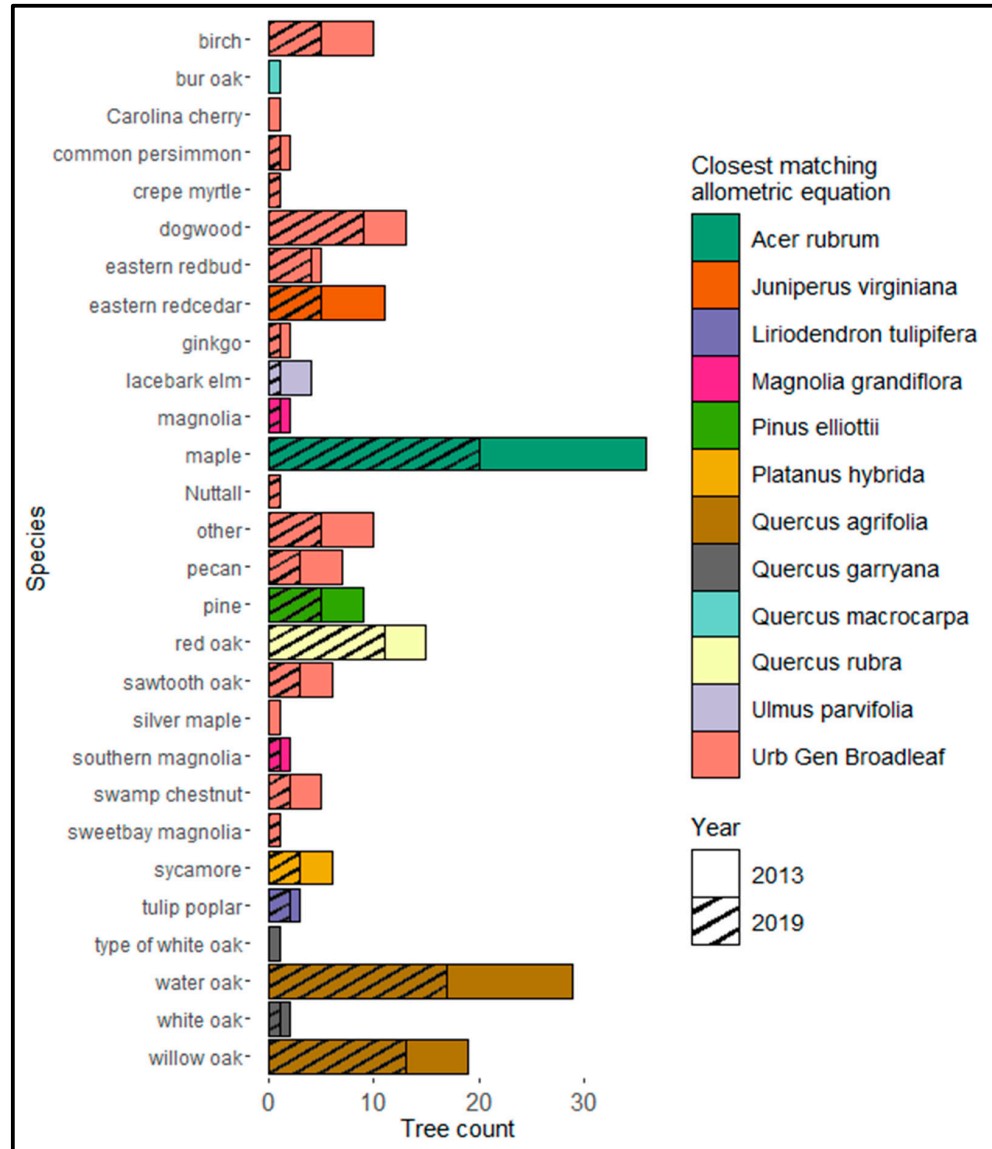

**Figure 5.** Tree species distribution and the matching species equation for volume or biomass calculation for the final 2019 (**stripes**) and 2013 (**no stripes**) datasets used for modeling, collected in Athens–Clarke County, Georgia, US.

**Table 4.** Descriptive statistics for the crown polygons for the final 2013 and 2019 isolated tree datasets, created from the *silva2016* algorithm in the lidR package.

|  | 2013 | 2019 |
| :---: | :---: | :---: |
| **Min, Max** | 1.0, 9.9 | 1.1, 10.6 |
| **Average** | 4.1 | 4.7 |
| **Standard deviation** | 2.3 | 2.0 |

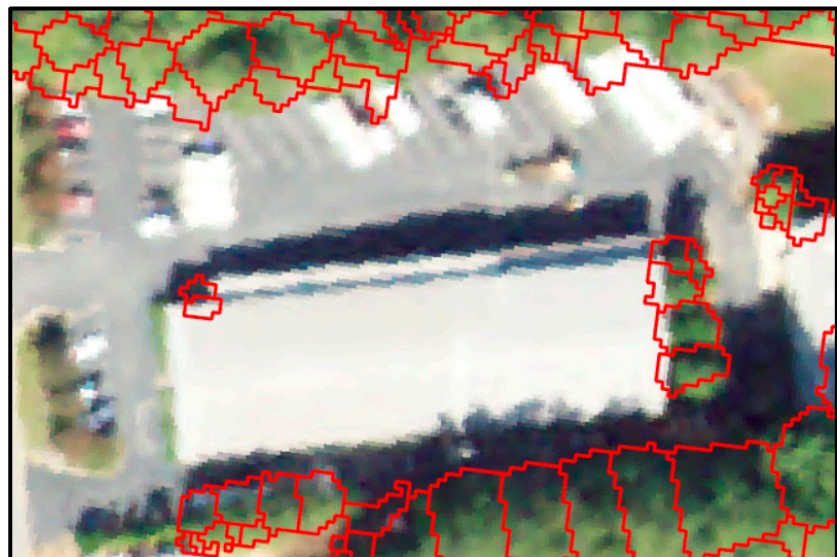

**Figure 6.** Crowns produced from *silva2016*, which show overflow over buildings despite filtering efforts. Data were collected in Athens–Clarke County, Georgia, US.

*3.3. Biomass Lasso Regression*

3.3.1. Results of 2013

The sample was divided into two groups based on the surrounding environment: woody and isolated. The residuals for both indicated non-normal distributions were analyzed. Following the Box–Cox results ($\lambda$ = 0.02), the two datasets were transformed using the natural logarithm. The results for the woody dataset had an adjusted $R^2$ = 0.44 ($n$ = 189), but the resulting final lasso regression for the isolated dataset was significantly better, with an adjusted $R^2$ = 0.85 ($n$ = 89). The RMSE was 1797 kg and the mean error (bias) was 225 kg. Table 5 shows the results of the 10-fold cross-validation results as compared to the final best model results.

**Table 5.** Results from the best 2013 model compared with averaged results from the 10-fold cross-validation, based on the isolated tree dataset collected in Athens–Clarke County, Georgia, US.

|  | Best Model | Cross-Validation |
|---|---|---|
| $R^2$ | 0.89 | 0.80 |
| Adjusted $R^2$ | 0.85 | 0.73 |
| RMSE (kg) | 1797 | 2009 |
| Bias (kg) | 225 | 209 |

The Shapiro–Wilk test revealed that the residuals were normally distributed for both woody and isolated tree datasets ($p$ = 0.25, 0.38, respectively). Using Cook's distance and examining the standardized residuals, outlying but non-influential tree points were removed. One was a crepe myrtle, which had over six stems and was therefore measured around the base and had a "large" diameter (52 inches) relative to its short height, throwing off the regression results.

Notably, 89 variables of over 300 variables were selected for the final lasso regression model based on the top 89 variables that had the highest correlation values to biomass above a threshold of r = 0.43. Of these 89 variables, 24 were selected using the best model, which had a lambda value of 0.01. The variables selected using the model are listed in Appendix A. Only two variables that used the aerial imagery and one of the vector-dependent variables (crown radius) were selected for the model.

Figure 7a shows the observed biomass to predicted biomass from the best-performing 2013 isolated tree sample model. The predictions were noticeably better when the biomass

was small—around 5000 kg or less (Figure 7b). Error increased with higher biomass values, with a slightly greater tendency toward underestimation.

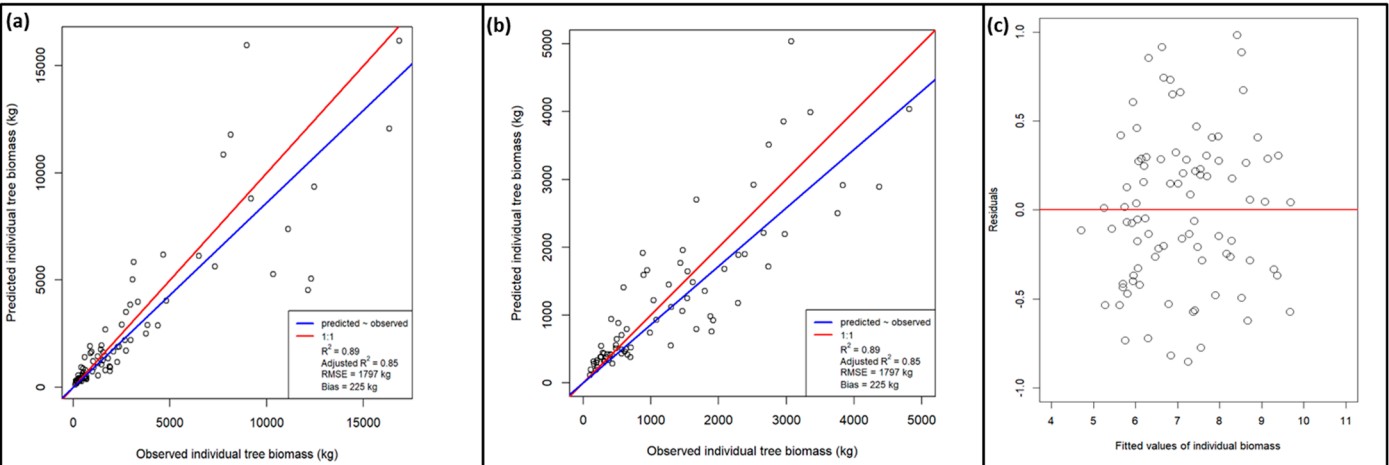

**Figure 7.** (**a**) Observed individual tree biomass to predicted individual tree biomass (kg) from the best-performing 2013 model from the isolated dataset (*n* = 89). (**b**) Observed individual tree biomass to the predicted individual tree biomass (kg) from the best−performing 2013 model from the isolated dataset for all trees less than 5000 kg. (**c**) Residuals versus transformed fitted individual biomass values for the 2013 model. Data were collected in Athens−Clarke County, Georgia, US.

### 3.3.2. Results of 2019

Similar to the 2013 results, the isolated dataset performed better than the woody dataset, and after removing the outlier crepe myrtle, filtering, and transforming, the results noticeably improved for both the woody and isolated datasets (*n* = 144, 116; adjusted $R^2$ = 0.67, 0.87). For the isolated model, the RMSE was 1444 kg, and the mean error (bias) was 194 kg. The Shapiro–Wilk test revealed that the residuals were normally distributed for the woody and isolated tree datasets (*p* = 0.66, 0.28, respectively). Other potential outlier trees were investigated but kept in the sample. A good example of one such tree was one downtown that had a noticeably higher diameter than its height, likely because of pruning on the side of the street and due to its powerline proximity. Such trees might appear to be potential anomalies in the sample but represent multiple urban tree situations.

Furthermore, 100 variables of over 300 variables were selected for the final lasso regression for the isolated dataset, based on the top 100 variables with the highest correlation values to biomass (r = 0.23). Of those, 27 were selected using the best model, which had a lambda value near 0. The variables selected using the model are listed alongside the 2013 results in Appendix A. Similar to the 2013 model results, only 3 of the 24 variables were from the aerial imagery. As with the 2013 model, the crown radius was chosen as a vector-dependent variable. As with the 2013 model, none of the variables included the NDVI, which is a common tree health metric. Table 6 shows the results of the 10-fold cross-validation results as compared with the final best model results.

**Table 6.** Results from the best 2019 model compared with averaged results from the 10-fold cross-validation, based on the isolated tree dataset collected in Athens–Clarke County, Georgia, US.

|  | Best Model | Cross-Validation |
| --- | --- | --- |
| $R^2$ | 0.90 | 0.81 |
| Adjusted $R^2$ | 0.87 | 0.77 |
| RMSE (kg) | 1444 | 2189 |
| Bias (kg) | 194 | 147 |

Figure 8 shows the observed biomass to the predicted biomass from the best-performing 2019 isolated tree sample model. Again, the predictions were better when the biomass was small. Error increased with high biomass values but did not show consistent over- or underestimation.

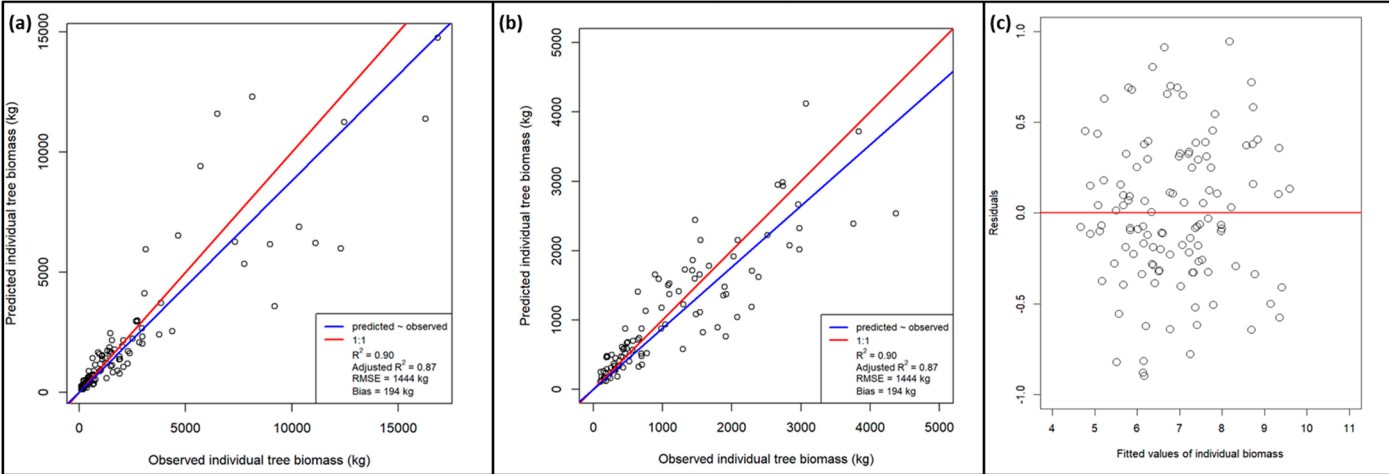

**Figure 8.** (**a**) Observed individual tree biomass to the predicted individual tree biomass (kg) from the best-performing 2019 model from the isolated dataset (*n* = 116). (**b**) Observed individual tree biomass to the predicted individual tree biomass (kg) from the best−performing 2019 model from the isolated dataset for all trees less than 5000 kg. (**c**) Residuals versus transformed fitted individual biomass values for the 2019 model. Data were collected in Athens−Clarke County, Georgia, US.

### 3.4. Total Biomass and Carbon Storage Prediction Results across Athens–Clarke County

The canopy cover was estimated from the crowns created. The estimated canopy cover was 56.4%. The total biomass was predicted per crown for every detected tree in the county. Crowns that had extremely high prediction values (with biomass values of hundreds of thousands of kg) were removed, the majority of which were notably growing in woody conditions. Figure 9 shows the estimation of biomass per crown in a section of Athens–Clarke County. Biomass per hectare and canopy cover per hectare estimates for each census block group were calculated and mapped for comparison (Figures 10 and 11). While canopy cover and biomass estimates appear to be closely proportional, there are some notable exceptions. For example, the two southernmost census block groups that are classified as deciduous land cover and residential land use show higher values of biomass than canopy coverage, whereas the westernmost central census block group that is classified as deciduous and natural and undeveloped shows higher canopy cover than biomass.

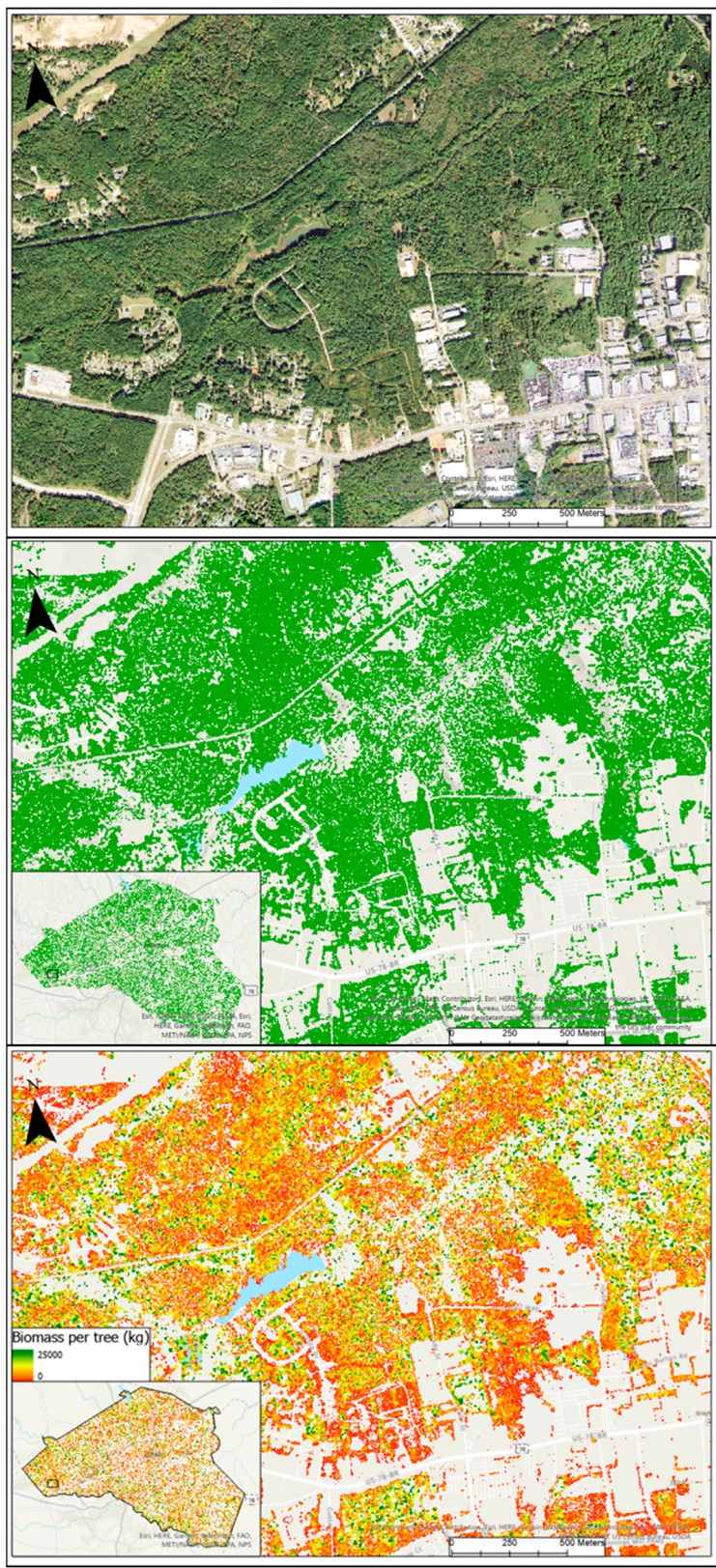

**Figure 9.** Aerial imagery of a section of Athens–Clarke County, GA (**top**), with its corresponding canopy cover (**middle**), and crowns of biomass (**bottom**).

**Biomass (metric tons) Compared to Canopy Cover per Census Block
Group, Classified by Predominant Land Cover Type**

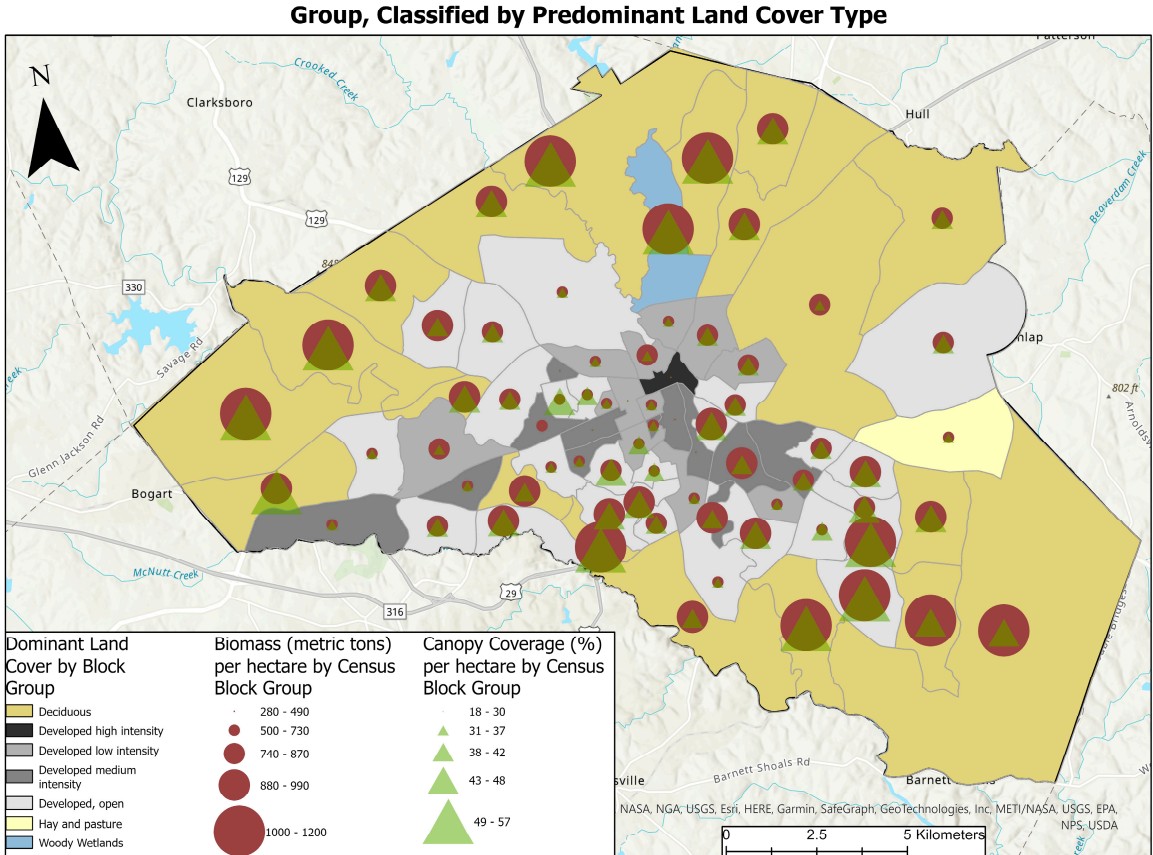

**Figure 10.** Canopy cover per hectare (green triangles) compared with biomass per hectare (brown circles) for each census block group in Athens−Clarke County, GA. Each block group is classified by its predominant land cover type as determined using 2019 NLCD data.

For land cover type, the biomass totals reveal that most biomass is stored in forests, with deciduous forests containing the most, followed by mixed forests and developed, open space (Table 7). In contrast, canopy-coverage totals show that most coverage is in deciduous forests, followed by "developed, open space" and "developed, low intensity". This contrast in canopy coverage compared with biomass is apparent in other land classes such as hay/pasture, which contributed 11.56% to canopy cover but only accounted for 4.36% of biomass. The biomass decreased along the urban spectrum from developed, open space to developed, high-intensity land cover.

**Biomass (metric tons) Compared to Canopy Cover per Census Block Group, Classified by Predominant Land Use Type**

**Figure 11.** Canopy cover per hectare (green triangles) compared with biomass per hectare (brown circles) for each census block group in Athens−Clarke County, GA. Each block group is classified by its predominant <u>land-use</u> type as determined using land-use data from Athens−Clarke County, GA.

**Table 7.** Distribution of canopy cover, total land area, biomass, carbon, carbon dioxide, and total contributions in Athens–Clarke County, Georgia, US, according to NLCD land cover type.

| Land Cover | Canopy Coverage (%) (Over All County Land, Forested and Non-Forested) | Canopy Cover Contribution to All Forested Land (%) | Total Land Area (%) (Forested and Non-Forested) | Total Biomass (Metric Tons, ±1.4 Error) | Total Carbon (Metric Tons, ±1.4 Error) | Total Carbon Dioxide (Metric Tons, ±1.4 Error) | Total Biomass, Carbon, and Carbon Dioxide Contribution (%) |
|---|---|---|---|---|---|---|---|
| Barren Land | 0.04 | 0.29 | 0.07 | 25,830.04 | 12,915.02 | 47,398.12 | 0.08 |
| Cultivated Crops | 0.01 | 0.25 | 0.01 | 3028.49 | 1514.25 | 5557.28 | 0.01 |
| Deciduous Forest | 19.32 | 23.53 | 34.22 | 11,025,858.68 | 5,512,929.34 | 20,232,450.68 | 33.29 |
| Developed, High Intensity | 0.25 | 2.81 | 0.44 | 135,592.04 | 67,796.02 | 248,811.39 | 0.41 |
| Developed, Low Intensity | 4.38 | 12.56 | 7.76 | 2,417,912.66 | 1,208,956.33 | 4,436,869.73 | 7.30 |
| Developed, Medium Intensity | 1.77 | 7.92 | 3.13 | 977,395.71 | 488,697.85 | 1,793,521.13 | 2.95 |
| Developed, Open Space | 8.71 | 16.09 | 15.43 | 4,947,701.30 | 2,473,850.65 | 9,079,031.89 | 14.94 |
| Emergent Herbaceous Wetlands | 0.07 | 0.17 | 0.12 | 49,316.13 | 24,658.06 | 90,495.10 | 0.15 |
| Evergreen Forest | 6.72 | 7.68 | 11.91 | 3,722,016.53 | 1,861,008.27 | 6,829,900.33 | 11.24 |
| Hay/Pasture | 2.53 | 11.56 | 4.49 | 1,444,816.29 | 722,408.15 | 2,651,237.89 | 4.36 |
| Herbaceous | 0.37 | 1.21 | 0.65 | 223,670.08 | 111,835.04 | 410,434.60 | 0.68 |
| Mixed Forest | 9.11 | 10.87 | 16.14 | 6,001,799.89 | 3,000,899.95 | 11,013,302.80 | 18.12 |
| Open Water | 0.21 | 0.87 | 0.38 | 139,710.75 | 69,855.38 | 256,369.23 | 0.42 |
| Shrub/Scrub | 0.37 | 1.02 | 0.66 | 252,277.22 | 126,138.61 | 462,928.71 | 0.76 |
| Woody Wetlands | 2.53 | 3.15 | 4.48 | 1,755,163.36 | 877,581.68 | 3,220,724.77 | 5.30 |

For land-use type, both biomass and canopy coverage followed the same ranking. The residential areas provided the majority of canopy cover and biomass, followed by natural and undeveloped, and then agricultural (Table 8).

**Table 8.** Distribution of canopy cover, biomass, carbon, carbon dioxide, and total contributions in Athens–Clarke County, Georgia, US, according to land-use type.

| Land Use | Canopy Coverage (%) (Over All County Land, Forested and Non-Forested) | Canopy Cover Contribution to All Forested Land (%) | Total Land Area (%) (Forested and Non-Forested) | Total Biomass (Metric Tons, ±1.4 Error) | Total Carbon (Metric Tons, ±1.4 Error) | Total Carbon Dioxide (Metric Tons, ±1.4 Error) | Total Biomass, Carbon, and Carbon Dioxide Contribution (%) |
|---|---|---|---|---|---|---|---|
| Agricultural | 4.65 | 8.24 | 9.73 | 2,611,232.31 | 1,305,616.15 | 4,791,611.28 | 8.35 |
| Commercial | 0.41 | 0.73 | 2.20 | 250,423.52 | 125,211.76 | 459,527.16 | 0.80 |
| Industrial | 1.03 | 1.83 | 2.79 | 580,650.04 | 290,325.02 | 1,065,492.82 | 1.86 |
| Mixed Use | 1.03 | 1.83 | 1.62 | 456,252.77 | 228,126.38 | 837,223.83 | 1.46 |
| Multifamily Residential | 1.83 | 3.24 | 3.87 | 1,061,655.79 | 530,827.90 | 1,948,138.38 | 3.39 |
| Office | 0.13 | 0.22 | 0.42 | 75,824.50 | 37,912.25 | 139,137.96 | 0.24 |
| Public | 1.70 | 3.01 | 3.89 | 874,054.56 | 437,027.28 | 1,603,890.11 | 2.79 |
| Recreation | 2.39 | 4.23 | 4.23 | 1,321,073.42 | 660,536.71 | 2,424,169.73 | 4.22 |
| Residential | 23.78 | 42.12 | 38.57 | 12,391,779.02 | 6,195,889.51 | 22,738,914.50 | 39.61 |
| Transportation, Communication, Utilities | 1.19 | 2.12 | 3.15 | 693,069.34 | 346,534.67 | 1,271,782.24 | 2.22 |
| Natural and Undeveloped | 15.83 | 28.04 | 20.54 | 9,258,574.03 | 4,629,287.02 | 16,989,483.35 | 29.59 |
| University | 0.20 | 0.35 | 0.72 | 90,258.27 | 45,129.14 | 165,623.93 | 0.29 |
| Roads | 2.28 | 4.04 | 8.45 | 1,623,346.89 | 811,673.44 | 2,978,841.54 | 5.19 |

## 4. Discussion

This research contributes to a growing body of urban tree biomass estimation from LiDAR data fusion in multiple ways. First, it shows how three-dimensional LiDAR data can be used to improve upon aerial photography-only biomass estimation methods such as in iTree Eco. Second, this is the first application, of which we are aware, of open source *silva2016* and *dalponte2016* crown segmentation algorithms to the urban environment. Third, in comparison to many other urban LiDAR biomass papers, this research shows how a non-species-specific relationship between biomass and LiDAR metrics and spectral variables can be modeled, which enables the application of the model to all other trees in the region. This provides a general approach that does not require knowing the species of each tree. Other urban studies have limited their datasets to limited sample sizes and only to select species, which hinders their models from being applicable on the larger city scale. Despite this, the model performance values still support previous individual tree biomass studies, which have similarly shown increased error with increasing biomass, generally in terms of more underestimation than overestimation [50–52]. For practical applications, users should be aware of the increasing error with higher biomass values, as well as potential underestimation from the model. Fourth, a comparison of biomass models from two LiDAR datasets of different densities shows that 1 point/m$^2$ actually performed rather well. This finding suggests that even with low density, cities can use low-density LiDAR data for biomass and carbon storage estimates. For research, however, greater density would allow for the creation of voxel-based metrics, which might significantly enhance model results [54,55]. More research should continue to investigate if low-density LiDAR data provides enough carbon estimation for the purposes of valuation and carbon stock. It is possible that the 2019 model's slightly increased performance was not due to higher point density but could be more attributed to its temporal proximity to field data collection and to the higher resolution 2019 aerial imagery. Fifth, experiences from this research will help provide technical guidance for future urban forest inventory practices.

Although a model for all species was developed, it was based on a subset of trees that were mostly isolated from other trees. Following the individual tree segmentation method, which was applied to the entire county, we applied our individual-tree-based predictive model. We acknowledge that while it is ideal to collect and incorporate field data from

woody conditions, matching the tree-level data to its counterpart in remote sensing data proved challenging. This is further complicated by the close tree proximity and complex vertical structure in woody conditions, making crown segmentation more difficult. A recent study further shows that most global navigation satellite systems (GNSSs) still fail to offer accurate enough tree point locations. This supports our own experience gathering tree point locations in the field [56]. The GPS point locations were frequently offset from the actual tree, which was possible to correct for the isolated tree dataset but much more difficult for the woody dataset. Highly accurate GPS data are needed to create individual-tree-level biomass estimates from woody conditions. The future use of a high-accuracy instrument such as a total station, although more time-consuming, would be needed to obtain proper point alignment with the tree crown. It is likely that the lack of observations from woody conditions led to the high overestimation of some crowns we subsequently removed, the majority of which were in woody conditions.

It is possible that modeling in the future could be subdivided into separate models. Because of the different growing conditions in an urban environment, it would be good to develop separate models for trees growing in woody conditions and those growing in isolated conditions. Based on our experience, it is challenging to gather accurate enough GPS data for trees growing in woody conditions, if the individual tree method is going to be applied. Currently, it is more likely that urban trees in woody conditions will still require plot-based biomass modeling methods, as is currently applied in traditional forestry [57]. In addition to those models, two other general models could also be explored: one for deciduous trees and one for evergreen trees. This could result in more accurate biomass estimates but would require the ability to accurately differentiate between the two types beforehand from remote sensing data. Using hyperspectral data, higher-density LiDAR data, seasonal LiDAR data, and wintertime imagery could make this possible. This again places more strain on governments and agencies to afford access to these types of data but can be considered. Further research can help pinpoint which tools offer the best differentiation between tree types for the future refinement of biomass models.

In addition to our suggestion that current studies focus on measuring isolated urban tree data for modeling, we further suggest that some alternative methods of measurement should be considered for trees with greater than six stems. Software such as iTree has set the standard for urban forest inventories that any tree with more than six stems should simply be measured one foot above the ground, around the base of the tree. This can pose problems during modeling, however, as our experience with one crepe myrtle showed. The diameter thus appeared large relative to its short tree height, confusing the model. Similarly, some urban trees were found to be potential outliers. These trees appeared to not follow the "typical" growth or dbh/height ratio due to cultural treatments. For example, we measured a tree of thick diameter but shorter than usual height, likely due to constant treatments as a result of its location downtown, beside a street, and underneath a powerline. Such cases will be present in urban environments and need to be taken into consideration.

As has been noted by many studies, individual-tree-level biomass modeling heavily relies upon the method of crown segmentation [58]. The crown area and radius variables can also be important in modeling, as there is a correlation between crown size and biomass. Our results support the finding that crown radius or diameter is important for modeling, as it was selected by both the 2013 and 2019 models [59]. This shows agreement with a 2020 study in which the crown diameter was found to explain the most variation in above-ground biomass in Eucalyptus as compared to all other variables used in the study [59]. Therefore, better crown segmentation should create more accurate crown variables for modeling biomass.

While land use and land cover distribution values followed a typical pattern (i.e., they showed that forested and agriculture types contributed the most forested land in the county, and values decreased as the land cover type became more developed), county maps revealed more difference between canopy cover values compared with biomass values. As noted above, some census block groups showed higher or lower values of

canopy cover than biomass distribution. While this could be due in part to the filtering of crowns with too high biomass estimates, which were mostly in woody conditions, these results could also indicate a trend in canopy coverage versus biomass in developing spaces. Specifically, these results indicate that developing urban spaces are managing to maintain canopy cover while reducing biomass overall. Biomass reduction is likely due to a few cultural practices: thinning understory growth in already existing woody areas and maintenance of trees in smaller, more restricted growing spaces. This illustrates the need for cities to consider both canopy cover and other metrics, such as biomass or leaf area, to more comprehensively estimate the distribution of tree benefits and services. Further investigation of the southernmost two census block groups that show more biomass than canopy cover reveals that these areas, despite their classification of residential and deciduous types, contain more open land and pasture. Increased biomass estimates might be due to edge effects from forest fragmentation, notably edges that, unlike in an urban environment, are not being heavily maintained and may affect model inputs.

## 5. Conclusions

The resulting 2013 and 2019 individual tree biomass models show that urban tree biomass estimation is possible using both 1 point/m$^2$ and 4 points/m$^2$ LiDAR data, fused with CNIR imagery. This is further possible despite having a lapse in time, in our case of 2013 remote sensing data and 2021 field tree inventory, as long as careful filtering of tree observations is applied. While the lower-density dataset still resulted in a highly adjusted R$^2$, the model that was based on higher-density LiDAR data showed increased performance. Because the inventory data used were trees that were mostly isolated from others, future research and model improvements should explore methods for incorporating urban trees that exist in woody conditions. An additional complication in the individual-tree modeling approach is the reliance on tree crown segmentation algorithms. While we found enhanced performance by detecting treetops using LiDAR data, proper crown delineation remains a challenge, especially in the urban environment with powerlines and buildings, which are not always completely filtered. Nevertheless, from the data fusion of LiDAR and imagery, individual-tree-level biomass models provide more detail than traditional urban biomass estimation methods, which have largely been imagery-based only.

**Author Contributions:** Conceptualization, A.P.; methodology, A.P.; software, K.A.H.; formal analysis, K.A.H.; investigation, K.A.H.; data curation, K.A.H.; writing—original draft preparation, K.A.H.; writing—review and editing, K.A.H. and A.P.; supervision, A.P.; project administration, A.P. All authors have read and agreed to the published version of the manuscript.

**Funding:** This research received funding from the University of Georgia School of Forestry and Natural Resources.

**Data Availability Statement:** Athens–Clarke County GIS and 2013 LiDAR data are available at: https://data-athensclarke.opendata.arcgis.com/. Accessed on 15 June 2021. The 2019 LiDAR data are available through USGS Earth Explorer at: https://earthexplorer.usgs.gov/. Accessed on 22 December 2021.

**Acknowledgments:** The authors would like to thank the field data collection crew for their hard work collecting field tree data.

**Conflicts of Interest:** The authors declare no conflict of interest.

## Appendix A

**Table A1.** Selected variables from the 2013 and 2019 lasso regression models, including spectral, LiDAR, and vector-based metrics. Ratios 1 and 6 were obtained from CNIR aerial imagery. The LII and height metrics were obtained from the LiDAR data. The crown radius was obtained from the crown shapefiles. The * indicates multiplication.

| Name | 2013 | 2019 |
|:---:|:---:|:---:|
| Ratio 1 | | X |
| NIR * DSM sd | X | X |
| Ratio 6 | X | X |
| Height mode above 1 m | X | X |
| Height median above 1 m | X | |
| Height standard deviation above 1 m | X | |
| Height variance above 1 m | | X |
| 5th percentile height above 1 m | X | |
| 10th percentile height above 1 m | | X |
| 15th percentile height above 1 m | X | |
| 45th percentile height above 1 m | X | |
| 95th percentile height above 1 m | | X |
| % 1st returns above 1 m | | X |
| % all returns above 1 m | X | |
| % 1st returns above mean (from points 1 m and above) | X | X |
| % all returns above mode (from points 1 m and above) | X | |
| Height mode above 1.8288 m | | X |
| Height variance above 1.8288 m | | X |
| Height kurtosis above 1.8288 m | | X |
| 5th percentile height above 1.8288 m | X | X |
| 10th percentile height above 1.8288 m | | X |
| 15th percentile height above 1.8288 m | X | |
| 45th percentile height above 1.8288 m | X | |
| 95th percentile height above 1.8288 m | | X |
| % 1st returns above 1.8288 m | X | |
| % all returns above 1.8288 m | X | |
| % all returns above mean (from points 1.8288 m and above) | X | X |
| Height mode above 14.6304 m | | X |
| Height variance above 14.6304 m | | X |
| Height kurtosis above 14.6304 m | | X |
| 5th percentile height above 14.6304 m | X | |
| 10th percentile height above 14.6304 m | | X |
| 45th percentile height above 14.6304 m | X | |
| 95th percentile height above 14.6304 m | | X |
| % all returns above 14.6304 m | X | |
| % 1st returns above mean (from points 14.6304 m and above) | | X |

**Table A1.** *Cont.*

| Name | 2013 | 2019 |
|---|---|---|
| % all returns above mean (from points 14.6304 m and above) | | X |
| % 1st returns above mode (from points 14.6304 m and above) | X | X |
| % all returns above mode (from points 14.6304 m and above) | | X |
| Laser Interception Index (LII) | X | |
| Crown radius (m) | X | X |
| Top of tree height (LiDAR) | X | X |

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
