# Peer review of "Biomass Estimation of Urban Forests Using LiDAR and High-Resolution Aerial Imagery in Athens–Clarke County, GA"

_forests, doi:10.3390/f14051064_

Round 1

Reviewer 1 Report

The manuscript presents various methods for estimating urban tree biomass from Lidar data fusion. While the topic is interesting, the overall quality of the paper is lacking. The format is poor, resulting in unclear text. The manuscript appears to have not been thoroughly reviewed, as evidenced by incomplete revisions in figures and tables, such as at lines 143-145 and lines 179-180. Additionally, the data acquisition and processing methods are not clearly presented, leading to confusion and uncertainty. Despite addressing a major challenge in biomass estimation, the paper fails to provide the necessary clarity for impactful results.

In my opinion, the manuscript requires significant reworking and thorough revisions to reach its full potential. Therefore, I recommend that the manuscript undergo major revisions before it can be considered for acceptance.

Some further considerations for improvement:

Lines 7-19: The abstract lacks organization. It is recommended to introduce Aims, Methods, Results, and Conclusions in a structured manner.

Lines 58-59: The relationship between 3D data, Lidar, and aerial or satellite imagery is not clear. The author should provide more specific explanations of their correlation for use in subsequent experiments.

Line 143: I recommend moving Figure 2 to Section 2.2 where it corresponds to the description of the picture.

Lines 179-180: The order and punctuation of the figures are incorrect. Please thoroughly review and correct the entire text.

Lines 120-121: Please explain the basis for selecting 9 m and the method for eliminating data. This paragraph lacks clarity in data acquisition and processing.

Lines 167-173: There is insufficient information on how the creation of three models was performed.

Lines 204-205, 215-219, 294-300: These statements must be explained and supported by similar examples from the literature or contrast tests.

Lines 315, 352: Please use proper punctuation and provide a clear description of what the title means.

Lines 327-328: It is not clear how the authors identified which trees need to be deleted. The basis for this processing or how it affects the regression results should be explained.

Lines 331-332: Please explain the selection process.

Lines 341, 345, 349, 378, 382, 386: Please label the units of the axes.

Lines 415-41: Please thoroughly check the order of the forms.

The paper heavily emphasizes "woody conditions" as a source of sample data, but what exactly constitutes "woody conditions" is not explained. Additionally, tree growth should be taken into account, as even trees of the same species can vary significantly in performance based on different growth years. The variability of individual trees is mentioned, but the difference in tree growth is not addressed.

Lastly, the full paper should include sections on Author Contributions, Funding, Data Availability, Conflicts of Interest, etc. It should also contain an index of images and references, and clarify any abbreviations used in the text for better readability.

The manuscript needs significant improvement in terms of clarity and flow.

The abstract is not well-organized and should follow a structure of Aims, Methods, Results, and Conclusions.

There is a lack of clear explanation regarding the relationship between 3D data, LiDAR, and aerial or satellite imagery.

Figures and tables, such as at lines 143-145 and lines 179-180, show that the paper was not fully revised before submission.

The order and punctuation of some figures are incorrect and need to be corrected.

The explanations regarding data acquisition and data processing are vague and need more details, such as the basis for selecting 9 m and the method of eliminating data.

There is insufficient information on how the creation of the three models was performed.

Reviewer 2 Report

Biomass estimation of urban forests is important for carbon storage study. However, the methods for estimating the biomass of individual trees using LiDAR data and images are quite common. So the novelty and innovation of the study should be further refined. There are some specific suggestions.

1.Line 58 “Limitations from image-only methods, as mentioned above” where mentioned the limitation above?

2. The field data were collected in the summer 2021, while the LiDAR data were collected in 2013 and 2019, how to consider the time difference between data.

3. In line 137 “A GPS was placed…” Please add the information of GPS, and the positioning accuracy under the tree canopy? In line 474-475 “Based on our experience, it is challenging to gather accurate enough GPS data for trees growing in woody conditions” So the positioning accuracy of individual tree under the tree canopy should be added. The accurate GPS data for trees growing in woody conditions could be obtained through total station.

4.Suggestion to add a table describe the field data, such as tree height, diameter at breast height, and crown diameter.

5. Line 147 “Four-band color near infrared (CNIR), 1- and 0.6-meter aerial imagery from 2013 and 2019 was acquired” How to deal with the difference in spatial resolutions between 2013 and 2019?

6. Line 154 “Machala and Zejdová (2014) [30]”, line 197 “Cieszewski et al. (2021)[37]” and “Taras and 197 Phillips (1978)[38]”. Is the style of references right?

7. Line 171-172 “To create the CHM, the OSAVI and NDVI values were extracted to each LiDAR point…” How are the images and LiDAR data registered, and what is the registration accuracy?

8. Line 182 “Using the filtered and normalized LiDAR data…” When did the LiDAR data undergo elevation normalization in the former?

9. Suggest moving the content of lines 233-241 after line 178.

10. Is the segmentation result of individual tree at the boundary of building in the middle of Figure 5, including some buildings, caused by the inaccurate registration of the image and LiDAR data? Because the segmentation results of individual tree are generated from the LiDAR data.

11. Line 304 “Figure 6 shows an…” The Figure 6 should be Figure 5.

12. Line 315 “3.3.12013. results” Suggestion to rephrase to “3.3.1 Results of 2013” The same suggestion to Line 352.

13. What is the average crown diameter of individual trees? How reliable is the individual tree segmentation results based on LiDAR data of 2013 with1 point per square meter?

14. The adjusted R2 of 2013 model is 0.85 and the adjusted R2 of 2019 model is 0.87. Is the difference between the 2013 and 2019 models because 2019 is closer to the 2021 field data collection, so the tree height and crown width extracted based on the 2019 LiDAR data are closer to the measured values in 2021, resulting in higher model accuracy rather than being caused by differences in point cloud density.

Moderate editing of English language

Reviewer 3 Report

The paper is devoted to the problem of the urban forest biomass estimation. Authors used LiDAR data, high-resolution images and in-situ measurements to build the biomass model. As a result of the study, the best variables were selected and the trees biomass was estimated with good accuracy. The topic of the paper is relevant, the advantage of the work is an attempt to construct an approach that does not require knowing the species of each tree.

I have a few questions and suggestions for the authors:

1.      In Abstract and beyond you say that other research with LiDAR was limited by area or species. But there are several global estimates of aboveground tree biomass, including those based on lidar data (Global Forest watch, ESA GlobBiomass, ESA Biomass Climate Change Initiative). Have you compared your results with these data?

2.      I also think that available products and methods for calculating AGB should be described in the introduction, just as you mentioned about LAI.

3.      Has an attempt been made to estimate the contribution of natural biomass growth to the error? Indeed, for 2013 data, this is 8 years before the time of field measurements in 2021, and for 2019 data, it is only 2 years.

4.      Please describe what software, programming languages and libraries you used at all stages of data processing.

5.      Figures 6-8 and 9-11 are very stretched and take up a lot of space, so it is difficult to compare them. Try to place figures of the same group (for example, 6-8) side by side, in one line.

6.      Paragraph headings 3.3.1 and 3.3.2 missing a space before the year

Round 2

Reviewer 1 Report

Line 151 and Line 198: There are discrepancies between the positions of the figures and their corresponding descriptions starting from Figure 2. Furthermore, there are two instances of Figure 2 in the text, and the punctuation is incorrect, which negatively impacts readability. It is highly recommended to properly index all images and charts in the manuscript and to thoroughly review the formatting and usage of symbols.

Lines 63-74: The author presents several professional terms, but fails to adequately explain their interrelationships. It is important to provide a more detailed explanation of the correlations between these terms.

Lines 218-219: The relationship between carbon and tree dry-weight biomass requires additional literature support for clarification.

Lines 294-300: The comparison requires either experimental evidence from previous literature or a comparison experiment to support its validity.

Figure 6 is missing from the text, while Figure 7 requires labeling of its subgraphs to differentiate them. Additionally, the last subgraph of Figure 7 does not include labels for the horizontal and vertical units.

Lines 338 and 354: There are two Table 3's in the text. Please ensure that the labels for the Tables are correct.

Line 409: Figure 8 is missing from the manuscript.

It is recommended to reorganize the layout of the Table by either splitting or consolidating data or by adjusting the typesetting.

Lines 435, 439, and 443: Figure 6, Figure 7, and Figure 8 are repeated in the manuscript.

The quality of the English language in the manuscript is okay. 

Author Response

Please see the attachment. Thank you for your time and patience!

Reviewer 2 Report

I would like to thank the author for providing answers to most of the questions. The quality of the article has been improved, but in the revised version, it still has some errors.

1.For the 11th question "Line 304: Figure 6 shows an example of crow overflow over buildings and the false detection of a few crows beside a roof. The "Figure 6 should be Figure 5 ”

The author’s response: We double-checked the text and Figures, and it appears correct. “Figure 6 shows an…” does correctly refer to its corresponding figure, which is the 6th figure in the manuscript.

I agree with the author said the figure is the 6th figure in the manuscript, but it should indeed be Figure 5 in the revised manuscript. And I also have doubts about the double-checked text and figures expressed by the author, as the 6th figure was written as “Figure 5. Crowns produced from silva2016 which show overflow over buildings despite filtering efforts. Data collected in Athens-Clarke County, Georgia, US.”, as shown in lines 342-343 of revised manuscript.

2.There are still many errors should be noted in the revised manuscript. For example, in line 151 and 198, both the figures are named with Figure 2, in line 244 and 315, both the tables are marked with Table 2, in line 338 and 354, both the tables are marked with Table 3. And Figure 5 on page 11 is followed by Figure 7 on page 12, while Figure 6 is on page 16.

3.Is the figure after line 409 missing?

Author Response

Please see the attachment. Thank you for your feedback and patience!

Reviewer 3 Report

Thank you for good work with paper!

Author Response

The authors thank reviewer 3 for their feedback and patience!